# Light sheet microscopy with acoustic sample confinement

Zhengyi Yang[1,9], Katy L.H. Cole[2], Yongqiang Qiu [3,10], Ildikó M.L. Somorjai [4,5], Philip Wijesinghe [1,6,7], Jonathan Nylk [1], Sandy Cochran[3], Gabriel C. Spalding [8], David A. Lyons [2] & Kishan Dholakia[1]

Contactless sample confinement would enable a whole host of new studies in developmental biology and neuroscience, in particular, when combined with long-term, wide-field optical imaging. To achieve this goal, we demonstrate a contactless acoustic gradient force trap for sample confinement in light sheet microscopy. Our approach allows the integration of real-time environmentally controlled experiments with wide-field low photo-toxic imaging, which we demonstrate on a variety of marine animal embryos and larvae. To illustrate the key advantages of our approach, we provide quantitative data for the dynamic response of the heartbeat of zebrafish larvae to verapamil and norepinephrine, which are known to affect cardiovascular function. Optical flow analysis allows us to explore the cardiac cycle of the zebrafish and determine the changes in contractile volume within the heart. Overcoming the restrictions of sample immobilisation and mounting can open up a broad range of studies, with real-time drug-based assays and biomechanical analyses.

[1] SUPA, School of Physics and Astronomy, University of St Andrews, St Andrews KY16 9SS, UK. [2] Centre for Discovery Brain Sciences, MS Society Centre for Translational Research, Euan MacDonald Centre for Motor Neurone Disease Research, University of Edinburgh, Edinburgh EH16 4SB, UK. [3] School of Engineering, University of Glasgow, Glasgow G12 8QQ, UK. [4] The Scottish Oceans Institute, University of St Andrews, St Andrews KY16 8LB, UK. [5] Biomedical Sciences Research Complex, North Haugh, University of St Andrews, St Andrews KY16 9ST, UK. [6] BRITElab, Harry Perkins Institute of Medical Research, QEII Medical Centre, Nedlands and Centre for Medical Research, The University of Western Australia, Perth, WA 6009, Australia. [7] Department of Electrical, Electronic & Computer Engineering, School of Engineering, The University of Western Australia, Perth, WA 6009, Australia. [8] Department of Physics, Illinois Wesleyan University, Bloomington, IL 61701, USA. [9] Present address: Electron Bio-Imaging Centre, Diamond Light Source, Harwell Science and Innovation Campus, Didcot OX11 0DE, UK. [10] Present address: Faculty of Engineering and Technology, Liverpool John Moores University, Liverpool L3 3AF, UK. Correspondence and requests for materials should be addressed to Z.Y. (email: zhengyiyang.phy@gmail.com) or to K.D. (email: kd1@st-andrews.ac.uk)

The field of biomedicine is seeing an increased use of model organisms that can inform us about the onset and progression of the disease. In particular, the use of zebrafish embryos has gained prominence in many areas or research, including studies of cardiovascular disease and in neuroscience. In parallel, there is an urgent need for the development of fast drug-based assays for biomedical analysis. A major advance for realising such studies would be contactless sample suspension combined with rapid and minimally photo-toxic wide-field imaging. This would inherently maintain the integrity of the natural physiological development of the specimen, which may be adversely affected when using agarose or other gels for specimen embedding[1–3].

For rapid wide-field imaging, light sheet fluorescence microscopy[4] (LSFM) is becoming established for the advantages it brings to such longitudinal imaging studies. The orthogonal geometry of the illumination and detection axes ensures high-contrast images, even when the detection numerical aperture (NA) is low. This imaging approach has the ability to minimise sample photo-bleaching and photo-damage[4]. As a result, LSFM has seen exceptional uptake for studies with large samples, such as whole zebrafish and intact cleared mouse brain, in the fields of developmental biology and neuroscience[5–8]. However, the standard procedure for LSFM is to completely surround the sample with agarose gel[9,10], often extending out to the walls of the sample chamber. This approach offers facile translation and rotation of the sample within the imaging system. However, the use of agarose or similar gels have drawbacks for longitudinal imaging since they dramatically reduce the diffusion rate of liquid interacting with the sample, restricting both the introduction of fresh media as well as the discharge of accumulated waste[11] (see Supplementary Note 1). Additionally, there are a number of organisms that exhibit abnormal development and behaviour when mechanically confined by embedding in the gel. For example, Aptasia is a sea anemone model organism where agarose gel embedding disrupts the dynamic host–symbiont relationship[1]. This key bottleneck can restrict detailed studies of dynamic processes, therefore, precluding real-time analyses. A major advance would be a method by which samples could be immobilised in their native environment in the absence of agarose or similar methods of restriction, thereby ensuring more realistic and informative biomedical studies. This would enable rapid access of experimental agents in the media to the sample and open new routes for real-time drug assays. Furthermore, a broad host of biomechanical analyses, unimpeded and unchanged by physical confinement, would become tractable[12].

A powerful route would be to realise a truly non-contact approach devoid of any mechanical obstacles in proximity to the sample. In principle, optical traps can be used for this purpose[13]. However, optical forces are notoriously weak, typically on the order of a few picoNewtons, such that trapping and translating samples larger than a few tens of microns in diameter becomes intractable[13–16]. A more powerful route to address this challenge is acoustic trapping, which has been demonstrated as an effective alternative to confine biological samples safely[17–21]. Specifically, the use of ultrasonic transducers for acoustic trapping offers four key advantages over its optical counterpart. Firstly, the wavelength of ultrasound in the fluid is orders of magnitude longer than that of light. This gives ultrasound an advantage in exerting strong forces on targets of the size from tens of microns up to several millimetres[22,23]. Secondly, ultrasound is able to carry much higher energy at a much lower wave speed and, hence, is easily able to impart forces on the order of micro-Newtons, leading to more stable trapping[24,25]. Thirdly, as the power is distributed over a larger region, the potential for damage caused by an acoustic trap is greatly reduced, compared to optical trapping[20,21]. Finally, ultrasonic transducers are easily miniaturised and inexpensive, hence, are significantly easier to integrate into existing wide-field imaging setups[26,27].

Here, we demonstrate the integration of an acoustic trap into a compact LSFM, to provide contactless sample confinement of zebrafish, amphioxus embryos, and ascidian embryos and larvae. To demonstrate the versatility of our approach, we perform a drug study on the application of verapamil and norepinephrine to 2-days-post-fertilisation (dpf) zebrafish larvae, and monitor the heartbeat response with light sheet imaging. We use optical flow analysis to track the dynamics of the zebrafish heart and quantify contractility. Such studies open up the prospect of original avenues in cardiovascular research, long-term imaging and the assessment of dynamic biological processes.

## Results

**Acoustic trapping.** Figure 1a (see Methods) illustrates our sample chamber design, which integrates two counter-propagating, concave, spherical bowl-shaped ultrasonic transducers into a Perspex chamber. Ultrasound waves emitted from the transducers form a standing wave in the volume of the chamber, and it is the resulting gradients in the acoustic field which act to trap the specimen. The counter-propagating geometry minimises the acoustic streaming effect on the axis of the beams[28,29]. With the sample thus immobilized, volumetric image stacks could be obtained via a customized LSFM setup featuring a synchronized scanning mirror in the illumination pathway and an electrically tunable lens (ETL) in the detection pathway (see Methods).

For an initial characterization, we examined the system's ability to trap test samples, including 100 and 500 μm diameter polystyrene and glass spheres. Trapping of the 500 μm glass spheres demonstrated a lateral acoustic force of at least 5 μN, which was determined from the balance of the acoustic force with gravitational and buoyancy forces also acting upon the sphere in water. Notably, this is many orders of magnitude higher than an optical trap can provide; for example, even in a counter-propagating beam geometry, which has been the most favourable for optical trapping of 'large' objects, forces of only 135 pN were reported by Thalhammer et al. with a considerable laser power of 500 mW and an associated sample heating of 10 °C[14].

**Fast scanning and detection.** The optical and acoustic design merits consideration. Conventional three-dimensional (3D) image stack construction involves translating the trapped sample along the detection axis. At the largest displacements, the microscope objective may impinge on the acoustic field. To eliminate this possibility, our design exploits an inertia-free scanning system to enable the acquisition of volumetric image stacks without moving either the microscope objective or the specimen. A synchronised scanning mirror and an ETL are used to scan both the light-sheet and the focal plane of the detection lens[30,31] through the body of the sample. As well as maintaining a stationary lens, specimen and stable acoustic field, this approach also allows 3D image stacks to be acquired at rates of tens of Hz, which is particularly advantageous for imaging active, dynamic samples.

Furthermore, we found that by setting the exposure time on the camera to match the period of one axial scan cycle, we could achieve acquisition of an integrated axial projection of the 3D image stack, i.e. integrating volume stacks into one single image. Images thus obtained are equivalent to an average intensity projection of the 3D image stack. By acquiring the image in this (optional) projection mode, the amount of data generated is dramatically reduced. So long as depth information of individual

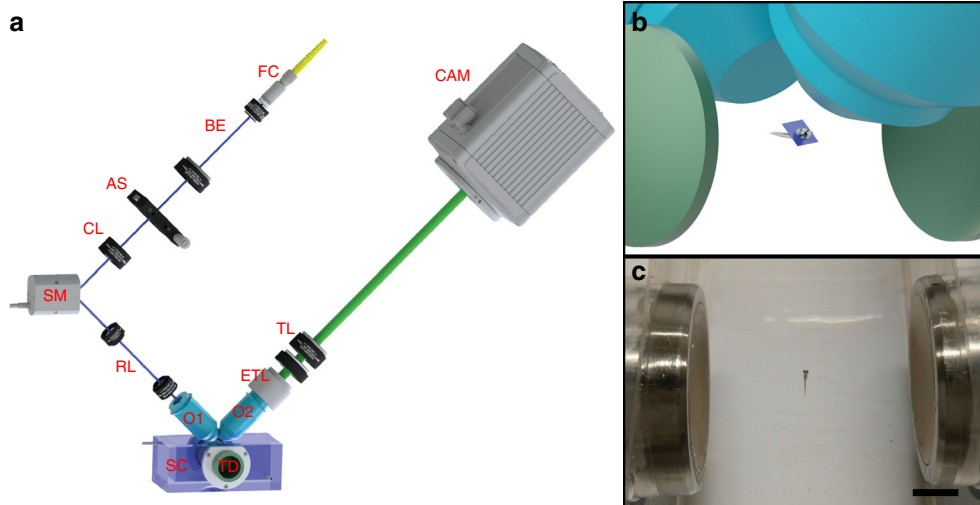

**Fig. 1** Light sheet fluorescence microscopy setup with an acoustic trapping chamber. **a** Schematic of the setup. The light sheet is scanned along the detection axis by the scanning mirror (SM) whilst the detection plane is synchronized with the light sheet by an electrically tunable lens (ETL). The acoustic sample chamber (SC) with acoustic transducers (TD) holds the sample while the images are taken. FC fibre collimator, BE beam expander, AS adjustable slit, CL cylindrical lens, RL relay lenses, O1 & O2 objectives, TL tube lens, CAM camera. **b** Schematic showing the acoustic transducers, trapped sample, the light sheet and the objective lenses. **c** Picture showing an acoustically trapped 5-days-post-fertilization (dpf) zebrafish larva. Scale bar denotes 5 mm

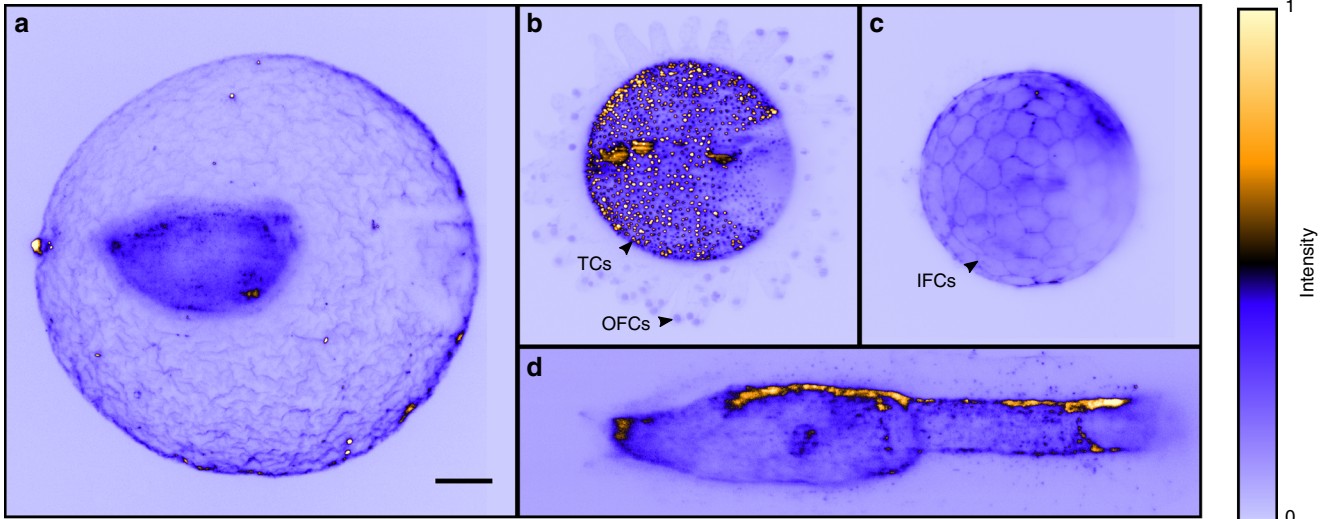

**Fig. 2** Light sheet fluorescent images of marine animal embryos and larvae. **a** Amphioxus early gastrula stage embryo with the chorion. **b** Ascidian embryo showing the nuclei, including at the tip of outer follicular cells (OFCs) and the monolayer test cells (TCs). **c** Ascidian embryo showing the cell membrane of inner follicular cells (IFCs). **d** Ascidian larva head with part of the tail (anterior to the left). Samples in figure (**a**, **c**, **d**) were stained with Alexa Fluor 555 conjugated WGA, and the images were obtained directly with long exposure time to achieve the projection effect. The sample in figure (**b**) was stained with SYTO 81, and the image was obtained by applying a maximum intensity projection to the image stack which contains individual sections. The image stack of (**b**) can be viewed in Supplementary Movie 7. Image intensities are normalized, colour bar is shown. Scale bar 50 μm applies to all panels

features is not required, this can be particularly useful for longitudinal imaging studies.

**Imaging of marine animal embryos and larvae.** Here, amphioxus (*Branchiostoma lanceolatum*) and ascidian (*Ciona intestinalis*) embryos, which are close marine invertebrate relatives of vertebrates, were acoustically trapped and studied longitudinally via LSFM. In contrast to what is commonly seen with the use of agarose gel, the original three-dimensional morphology of the samples, in particular, the chorion (fertilization envelope), was kept intact under acoustic positioning (Fig. 2a and Supplementary Movie 1). In addition, we were able to observe the

movement of the chorion with time-lapse LSFM imaging in mid-gastrula stage embryos (see Supplementary Movie 2). Normally the embryo begins to rotate within the chorion due to the development of ectodermal cilia[32]. In our system, the opposite was observed; the embryos maintained a fixed orientation whilst the chorion was seen to rotate. We attribute this to the acoustic gradient trap which holds the embryo in a fixed, energetically favourable, orientation whilst the ciliary motion causes the rotation of the chorion around the embryo. We also observed that the rotation speed increased with the age of the embryos, particularly just prior to hatching (mid-neurula stage) (see Supplementary Movie 3). When multiple embryos were trapped in the same acoustic gradient potential energy well, the chorions spun in

different directions (see Supplementary Movie 4). Numerous puncta can also be seen moving in the extracellular space between the embryos and the fertilization envelope. To the best of our knowledge, these have not been characterized before in amphioxus, and we believe may represent extracellular vesicles or exosomes[33].

In our images of ascidian embryos, as in Fig. 2b, we were able to monitor sub-cellular structures such as the nuclei located at the tip of outer follicular cells (OFCs), as well as the large vacuole structure in the follicular extension. The monolayer test cells (TCs) which form the epithelial layer were clearly resolved. Using time-sequence imaging, we tracked the disappearance of TCs at around 4 h post-fertilization, possibly representing apoptosis of these TCs[34]. Figure 2c shows the outline of inner follicular cells (IFCs), which were stained with a membrane-staining agent Alexa Fluor 555 conjugated Wheat Germ Agglutinin (WGA). Figure 2d shows an ascidian larva. Although the larva twitched its body, utilizing developed muscles to try to move beyond the trap, it was nevertheless confined in the trap throughout our LSFM experiment. Capturing such rapid movements demands high-speed imaging. We were able to capture 3D image stacks in 100 ms or less, enabling us to visualise such movements due to our inertia-free scanning design.

**Dynamic response to drug treatment.** Whilst acoustic trapping provides suitable force for the confinement of micro-organisms, we demonstrate that it can also suspend aquarium fish such as zebrafish larvae, with the presence of low-dose anaesthetics. Currently, suspension of the specimen in LSFM is commonly achieved through the use of agarose gel. The compound diffusion speed in agarose gel is dramatically reduced by the quasi-rigid fibre structure of the agarose (see Supplementary Note 1), hence limiting the efficacy of drug treatment studies. With acoustic trapping, the sample is held directly in the optimal medium, hence, drug delivery is straightforward. To demonstrate our system's suitability for real-time drug assays, we trapped 2-dpf zebrafish larvae and monitored their heart rate via LSFM, using the projection imaging mode as described above.

Responses of the heart rate were recorded before, during and after treatment with verapamil which is a calcium channel blocker used to treat high blood pressure and to decrease heart contraction. During the whole process, 80 mL L$^{-1}$ tricaine was used in order to prevent attempts to swim away and stress.

The dynamics of the zebrafish heart were quantified using optical flow analysis (see Methods). Optical flow analysis was used to map the local 2D displacement field in the heart between each consecutive frame (i.e. velocity) based on the change in the spatial and temporal image intensity gradients[35]. Figure 3 shows select 2D vector plots of velocities in the heart across the cardiac cycle. The contraction of the ventricle can be seen from the convergence of the velocity vectors towards its centre. Similarly, the relaxation of the ventricle and the contraction of the atrium can be seen from the respective divergence and convergence of the velocity vectors, concentrated around certain regions of the heart. The full time-lapse of deformation can be accessed in Supplementary Movie 5. To quantitatively profile, the mechanical deformation of the heart, the local volumetric strain rate was calculated from the estimated velocities (Methods). Figure 4a shows the strain rate mapped from the velocities in Fig. 3a, representing the spatially resolved relative change in volume between image frames. The full time-lapse is provided in Supplementary Movie 6. The strain rate can be integrated across the entire set of images, producing the total volumetric strain, i.e.

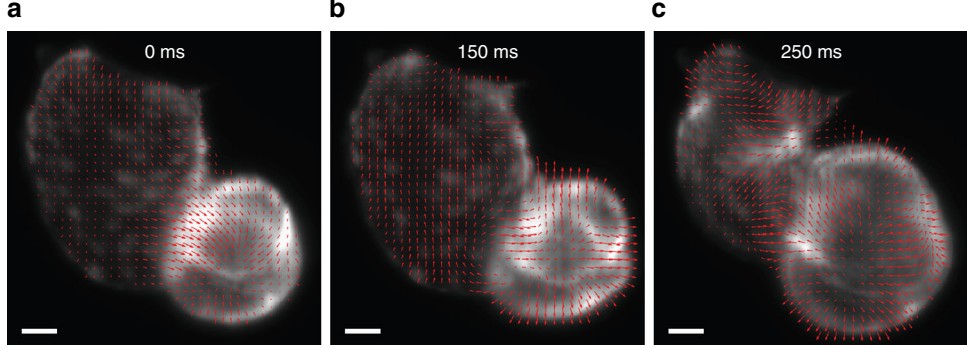

**Fig. 3** Velocity vector plot of the zebrafish heart estimated using optical flow analysis illustrating stages of the cardiac cycle: (**a**) contraction of the ventricle; (**b**) relaxation of the ventricle; (**c**) contraction of the atrium. Time elapsed relative to the first frame is noted on the figures. Scale bar is 50 μm (see also Supplementary Movie 5)

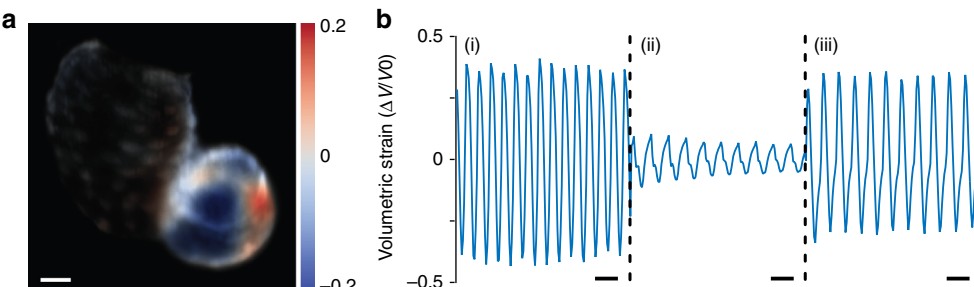

**Fig. 4** Contractility mapping of the zebrafish heart. **a** Volumetric strain rate (change in volume between frames) corresponding to Fig. 3a. Scale bar is 50 μm. **b** Selected traces of the total volumetric strain of the ventricle (i) before the drug is added (20 min), (ii) after the drug is added (50 min), and (iii) after the drug is washed away (150 min). The scale bar denotes 1 s (see also Supplementary Movie 6)

the total change in the volume of the heart (strength of contraction). Figure 4b shows selected plots of the strain in the ventricle throughout the experiment. Prior to the addition of verapamil, the peak-to-peak strain amplitude of close to 100% suggests that the ventricle expands and contracts by a volume equivalent to its resting size, while after the addition of verapamil, the volume of contraction is significantly lower (20–30% of its resting size). Normal ventricular contraction is significantly recovered after the removal of the drug. Verapamil also induced a slower heartbeat, as evidenced by the longer period in Fig. 4b (ii), which returned to normal after removal of verapamil.

The three stages of this experiment are indicated by vertical lines in Fig. 5. During the first stage, a baseline heart rate was determined, prior to the introduction of verapamil. In stage two, 40 mg L$^{-1}$ verapamil was introduced, and our data reveals that the heart rate of the zebrafish dropped as much as 48%. We have independently verified that it took ~2 min for the drug to be delivered to the location of the fish (see Supplementary Note 1), yet our data reveals that, it took ~5 min for the heart rate to start decreasing. Clearly, in addition to the time required for drug delivery within the medium, time is also required for diffusion into the sample. At stage three, the active compound was washed away with fresh medium, and the heart rate partially recovered, although only one sample returned to its original rate. Six out of eight tested samples are shown in Fig. 5. Of the two rejected samples, one lost rhythmic beating after drug application, whilst the heart rate change of the other was less than 10%. A control experiment was performed under identical conditions, but

without the addition of any drug; in that case, heart rates stayed within 100 ± 2.77% $\Delta f/f$ over a 2-h period of LSFM monitoring.

Figure 5 shows the peak-to-peak amplitude of the volumetric strain and the heart rate across the entire experiment and all samples. The heart rate and amplitude were recovered through Fourier analysis (Methods). The data was averaged across central portions of the atrium and the ventricle.

The beat amplitude follows a similar trend. The amplitude is normalised to the resting volume of the heart; however, there is significant variability between each animal. Upon the addition of verapamil, the amplitude of contraction decreases with an exponential decay, dropping from around 50% to 30% of volume contraction. With the removal of the drug, the amplitude partially recovered, still exhibiting significant inter-sample variability.

Additionally, the experiments were repeated with 1 mM norepinephrine (NE) to illustrate increases in heart rate. Figure 5b shows the normalised heart rate from five 2-dpf zebrafish. Upon addition of NE, there is a gradual and consistent increase in heart rate to around 7%, and a similar drop to resting heart rate after drug wash-out. When repeated on two 3-dpf zebrafish, the heart rate increase was 17% (Supplementary Note 2), consistent with the previous studies[36]. Interestingly, the sensitivity of zebrafish to adrenergic agonists greatly increases in the early stages of development. For instance, the expression of adrenergic receptor genes in 3–4-dpf zebrafish can be 2-fold larger than in 2-dpf zebrafish[37].

Similar heart rate tests were carried out on 5-dpf zebrafish larvae, which were treated with 10 mg L$^{-1}$ verapamil, and on 4-

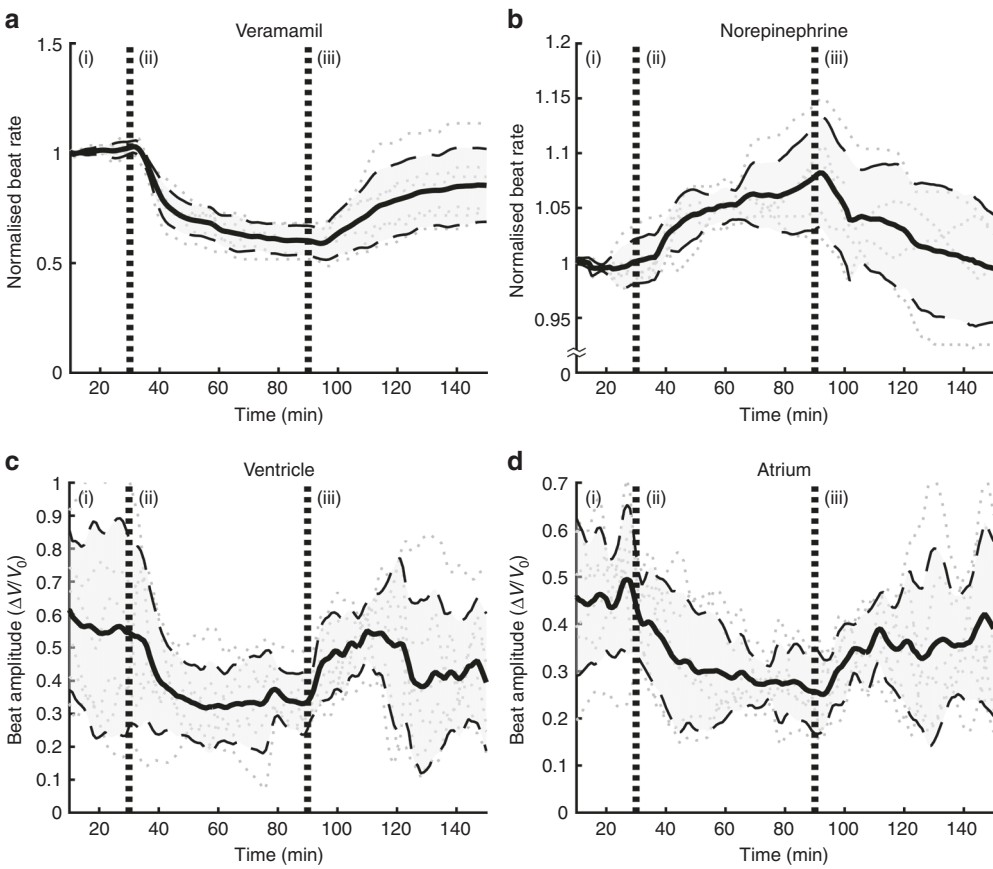

**Fig. 5** Zebrafish heart rate and beat amplitude with the addition of drugs. Zebrafish heart rate change due to (**a**) verapamil and (**b**) norepinephrine, normalised to the resting rate. Contractile beat amplitude in the ventricle (**c**) and atrium (**d**), upon addition of verapamil, presented as the total peak-to-peak amplitude of the volumetric strain. Each plot is presented as a mean value (solid lines) with error bounds of one standard deviation (dashed lines) across all results (dotted lines). No drug was added in period (i) for 30 min. The drug is added at beginning of period (ii), then washed away in period (iii). Sample size: 6 (**a**, **c**, **d**) and 5 (**b**)

dpf larvae, which were treated with a high-dose of tricaine (see Supplementary Note 2). Interestingly, following the expected decrease in heart rate from verapamil, in both experiments, after drug removal, the heart rate returned to the original levels observed prior to treatment. This may be due to the lower verapamil concentration or increased drug resistance in these more mature samples compared to that of younger larvae. In the reported data, we also see that tricaine became effective quickly, whilst obvious changes in heart rate did not occur until 30 min after verapamil was applied. Such differences are likely due to a combination of variability in drug diffusion rates into the sample and differences in the pharmacological mechanisms for each drug.

Viability checks were performed by acoustically trapping pacified 4-dpf larvae continuously for 16 h: no difference between the heart rate before and after trapping was detected. In addition, a 2-dpf larva was trapped continuously for 56 h. At the end of the trapping experiment, the animal showed no aberrant behaviour except a slightly reduced heart rate which may be attributed to long-term exposure to tricaine. Taken together, these control experiments demonstrate that acoustic trapping in and of itself has little to no detrimental effect on the biological samples studied here, even after prolonged exposure.

## Discussion

Agarose is currently in widespread use as a method for embedding samples during imaging studies, in particular, those utilising LSFM; however, the means of physical trapping constrains potentially alters the development and physiology of the organism[1–3]. Here, by integrating ultrasonic transducers into the sample chamber, we demonstrated contact-free suspension of the sample in optimum fluid. Compared to the use of optical forces, acoustic trapping is particularly advantageous for large samples, as stable confinement is achieved when the wavelength of the ultrasound is commensurate with the dimensions of the sample. We have demonstrated trapping of samples ranging from hundreds of microns to several millimetres in diameter, utilizing MHz frequency transducers. If needed, lower frequency transducers could be utilized for even larger specimens. Our design for a sample chamber with integrated acoustic trapping capability is independent of the optical setup, meaning that such an arrangement can easily be retrofitted into other LSFM setups.

The intensity of ultrasound required for trapping has been proven through a number of studies to cause little harm to biological samples. Some of those tests were performed on organisms over relatively short exposure times[17–20], whilst others were performed on mammalian cells with MPa ultrasound pressure over longer periods of time up to days[38–40]. Here, we have performed a long-term viability study with zebrafish larvae, as a step towards improving our understanding of the effects of long-term acoustic trapping upon a range of marine organisms. The maximum input voltage of 7.50 V was used, corresponding to acoustic pressure amplitude of 0.82 MPa and peak acoustic intensity of 22.58 W cm$^{-2}$. This is in line with other long-term cell viability studies in the literature[39,40], and the intensity is much less than that used for focused ultrasound therapy[41]. Our findings are that, even for prolonged acoustic exposures, no adverse effects have been observed.

Although often small in size and transparent with good optical qualities, many marine embryos and larvae represent particular challenges for live imaging. Premature removal of the chorion, prior to normal hatching, can affect the organisation of ectodermal cells or the expression of genes involved in left–right symmetry generation in ascidians[42,43]. Moreover, many embryos are covered by cilia (e.g. frogs, amphioxus, ascidians, sea urchins, sea anemones), which may not only be required during hatching but also to propel them through the fluid medium prior to overt muscle development. Methods to immobilise these specimens for imaging require physical constraint, for example between glass coverslips, or by the application of compounds that may disrupt ciliary function—both of which can perturb normal development. Our results underscore the potential of acoustic trapping as a method with promise for longitudinal imaging of marine and other small aquatic embryos, particularly if combined with the use of fluorescent reporters and transgenic lines, or with the application of compounds, demonstrated here for zebrafish.

There is ample evidence that mechanical cues, such as physical confinement, can alter biological processes[1–3,12]. Rapid estimation of displacements from high-throughput dynamic images coupled with the high spatio-temporal resolution of LSFM can reveal a broad range of these mechanical processes[12]. Optical elastography, the imaging of tissue mechanics using optics, is an emerging technique[44]. The combination of sub-micrometre resolution and rapid imaging, to date, has been challenging to achieve, which may see LSFM emerge as an advantageous tool in this field.

We have demonstrated a straightforward design for fast imaging of samples through acoustic suspension and LSFM imaging. In our initial design, the ETL was placed directly after the detection objective, enabling us to keep the size of the setup minimal whilst still achieving sub-micron resolution (Supplementary Note 3). An alternative position for the ETL, as well as dual-side light-sheet illumination, is included in a second design (see Supplementary Note 4). This dual-sided design may be more suitable for larger specimens.

Here, we have reported the response of zebrafish larvae to verapamil, norepinephrine and high-dose tricaine, monitored via analysis of LSFM time-series data. This allows heat rate frequency and amplitude analysis. This experimental method enabled us to place clearer constraints upon the dynamic response of the specimen to the compounds applied. Without the methods shown here, drug delivery is typically achieved by pumping the medium containing the drug compound into the sample chamber, which is filled with agarose or similar gel. Due to the resulting limitations on perfusion speed, the time required for the compound to reach the location of the sample is hard to quantify[11] (see Supplementary Note 1). In the absence of an embedding gel, fluid flow through the sample chamber and acoustic streaming effects may potentially aid standard diffusion, shortening required drug exposure times. The methods presented here should allow for the incorporation of further enhancements to drug delivery, offering still more precise timing measurements to be performed.

## Methods

**Optical setup**. Elements of the optical portion of our light sheet setup were adapted from the previous work[45]. Except for the lasers used, the optical setup was contained within the area of a $300 \times 400$ mm$^2$ breadboard (see Supplementary Note 5). The laser beams (Laser Quantum, finesse, 5 W, 532 nm, and M Squared frequency-doubled SolsTiS, 700 mW, 488 nm) were coupled into a single-mode fibre and sent onto the collimator on the breadboard. The beam was expanded and passed through an adjustable slit and a cylindrical lens to form a light sheet. This sheet was projected, by a pair of relay lenses, onto the back aperture of the illumination objective (O1, Olympus 10×/0.3, water dipping). The fluorescence signal was collected using a sCMOS camera (CAM, Hamamatsu Orca Flash 4) via an orthogonally mounted detection objective (O2, Olympus 20X×/0.5, water dipping). To eliminate any effects that the imaging objective might have upon the acoustic field when translating the sample, the sample was kept stationary; instead, the light sheet and detection plane were scanned, in synchronized fashion[30,31]. Displacement of the light sheet was achieved via a scanning mirror (SM, PI S-334.2SL1) located before the relay lens pair, and synchronized refocusing of the detection optics was achieved via an ETL (Optotune EL-10-30-C) (see Supplementary Note 6). MATLAB software was used to control the camera, function generator and ETL. Image stacks were constructed with open-source software FIJI[46]. The heart-beat analysis was performed with MATLAB.

**Acoustic setup**. For sample positioning, we created a counter-propagating dual-beam acoustic trap via two spherically-focused bowl-shaped transducers. The active material within the transducers was chosen to be PZ26 from Meggitt A/S. This commonly used piezoelectrically-hard piezoelectric material has a high $Q$ value and is able to provide large power and force. The outer diameter of the bowl-shaped active element was 20 mm with a focusing radius of 16 mm. This resulted in a $f$-number ($f\#$) of 0.8, providing a good compromise between the transducer physical size and the ultrasound focal region. The 16-mm focusing radius led to 32 mm axial length for the resonator, allowing space for dipping in the imaging objectives. The thickness of the active element was chosen to give a resonant frequency of ~1.5 MHz, resulting in 1 mm wavelength ($\lambda$) of ultrasound in water and 1.12 mm beam diameter, similar to the size of trapping targets. The use of small $f\#$ and high frequency also ensured a tight focus and high acoustic pressure at the focal point, leading to large pressure gradients and trapping forces near the trapping spot in the confocal system. The confocal system formed a quasi-standing wave field, where the dense objects were acoustically moved away from energy density maxima (pressure anti-node) and trapped at the energy density minima (pressure node). The likelihood of cavitation is largely suppressed by our selection of a MHz-range operating frequency[20], hence avoiding damage of the biological samples. Detailed transducer design and characterization can be found in Supplementary Notes 7 and 8. The active piezoelectric material was mounted into a stainless steel tube with 20-mm inner diameter, then conductive silver epoxy was applied between the front surface of the active element and the metal tube for electrical grounding. The transducers were air-backed to maximize the output power. Previous studies of acoustic traps with similar configuration have demonstrated stable trapping of targets ranging from 2.1 μm glass particles[47] to frog egg clusters (~1 mm)[28]. Our sample chamber is made of 10 mm-thick Perspex sheet with inner dimensions of $32 \times 30 \times 30$ mm$^3$ ($L \times W \times H$). This large chamber also minimises acoustic streaming at the target trapping site. Two holes were made in the side of the chamber, for the transducers, with O-rings around the transducers added to provide a water-tight seal. A function generator (LXI3390, Keithley) generated the alternating voltage used to drive the transducers at the resonant frequency of the system. A customized RF current amplifier is employed to ensure sufficient power for the transducers.

**Amphioxus embryo preparation**. Ripe *B. lanceolatum* adults were maintained at the Gatty Marine Laboratory (Scottish Oceans Institute) in a semi-closed recirculating seawater system at 16–17 °C prior to heat shock-induced spawning as per ref. [48]. Eggs and sperm were individually collected and mixed for controlled fertilization, and left to develop at 19 °C until the experiment. The embryos were incubated in 10 μg mL$^{-1}$ concentration of Alexa Fluor 555 conjugates (WGA, Invitrogen) at 20 °C for 1 h, followed by two washes with 0.22 μm-filtered seawater before being transferred to the acoustic trapping chamber for LSFM imaging.

**Ascidian embryo preparation**. *C. intestinalis* adults were collected from the west coast of Scotland, then maintained in the circulating seawater aquarium system at the Gatty Marine Laboratory (Scottish Oceans Institute) at ambient seawater temperature. Gametes were individually harvested from multiple ascidian adults; eggs were mixed with sperm for fertilization. Fertilized eggs were poured into a 40 μm cell strainer to remove excess sperm, then left to develop in filtered seawater at approximately 20 °C. Before imaging, the embryos were incubated in 5 μM SYTO 81 (Invitrogen) solution for staining for 30 min, followed with two washes to remove residual dye. The embryos were then transferred to the acoustic-trapping sample chamber for imaging.

**Zebrafish embryo preparation and drug treatment**. The zebrafish (*Danio rerio*) transgenic line (*mbp:GFP*)[49], which has *cmlc2:GFP* reporter expression in the heart was used to image heartbeat. Adult zebrafish, up to 2 years of age, were used for breeding purposes. All experimental analyses in the manuscript were carried out on animals up to 5 days post-fertilisation. All breeding and maintenance was carried out with approval from the UK Home Office and according to its regulations, under project licenses 70/8436. The relevant breeding and maintenance protocols were approved by the University of Edinburgh Institutional Animal Care and Use Committee.

Zebrafish embryos/larvae were maintained at 28 °C. The trapping chamber was filled with E3 medium at room temperature maintained at 22 °C. 80 mg L$^{-1}$ of tricaine was used as a baseline for the reported experiments, to help pacify the larvae. Zebrafish larva was transferred into the imaging chamber with a plastic pipette and acoustically trapped in the centre of the chamber.

For drug treatment, a peristaltic pump (minipulse 3, Gilson) was used to circulate fluid between the sample chamber and a beaker though polyvinyl chloride (PVC) tubing. For stage one, E3 medium with baseline tricaine was circulated between the sample chamber and the beaker in a closed loop. Starting at stage two, a high concentration of drug was added into the beaker, whilst the fluid was circulated in a closed loop, resulting in desired final concentration. For stage three, fresh medium with baseline tricaine was pumped into the sample chamber whilst the outlet was discharged to waste, so that the concentration of the drug decreased gradually. The pumping speed was 20 mL min$^{-1}$, and fluid volume in the sample chamber was 57.5 mL.

For heartbeat frequency analysis, image sequences were taken for 15 s every minute at a frame rate of 20 Hz. The peristaltic pump was stopped while image stacks were taken, to minimise the disturbance caused.

**Zebrafish sample size**. The number of zebrafish required per group (heart rate decrease, heart rate increase, control) was determined from the 'resource equation' method[50]. Based on 3 groups, 5/6 fish per group was sufficient.

Fish were randomly allocated to each group from the same population. Blinding was not relevant to this study as all imaging and image analysis was automated.

**Zebrafish heartbeat rate extraction**. Optical flow analysis was performed in MATLAB on consecutive LSFM intensity images. Optical flow was performed using the Horn–Schunck (HS) method[35], which estimates the image displacements by minimising an objective function comprising the local spatial and temporal image gradients (such as seen from LSFM image edges and features). It is further regularised to a global smoothness, i.e. a continuity of the displacement field gradients, to ensure a unique solution. The HS method is particularly suited to mechanical deformation problems, as a continuity in the displacement field gradients is likely. The major advantage of the optical flow method over the image correlation techniques, which are commonly used in tracking mechanical deformation, is its computational speed. The analysis performed in this study took 50 ms per frame, compared to digital image correlation, which took 2–5 min per frame. This speed enabled high-throughput analysis.

The deformation field of the heart, i.e. the strain tensor field, is given as: $\varepsilon = \frac{1}{2}\left(\nabla \mathbf{u} + (\nabla \mathbf{u})^{\mathrm{T}}\right)$, where $\mathbf{u}$ is the displacement vector field. Volumetric strain (the change in volume over the initial volume of an object) is the trace of the strain tensor: $\Delta V/V_0 = \mathrm{tr}(\varepsilon) = \varepsilon_{11} + \varepsilon_{22} + \varepsilon_{33}$, or simply the divergence of displacement: $\Delta V/V_0 = \nabla \cdot \mathbf{u}$. Volumetric strain between each frame was calculated and smoothed using a Gaussian kernel with an 11-μm standard deviation, weighted by the strength of the image gradients (related to the quality of optical flow estimation). Since the analysis was performed between each frame, the estimated values were velocities and strain rates (1/frame). Thus, the total volumetric strain was calculated by integrating across frames.

For each image sequence, the heartbeat rate and the volumetric strain amplitude were extracted for regions in the atrium and ventricle using Fourier analysis. The power spectral density (PSD) was calculated for temporal measurements in each location. The frequency peak of the PSD is the beat rate, whilst the amplitude of the PSD at the peak, and its harmonics, were related to the amplitude of contraction[51]. Such analysis discards the contributions of other noise sources not related to the beat.

**Code availability**. The code used for the analysis of data in this study is available in GitHub with the identifier: https://github.com/philipwijesinghe/Acoustic-trap-LSFM.

## Data availability

All data supporting the findings of this study are available in the University of St Andrews Research Portal with the identifier https://doi.org/10.17630/cc36c86a-5ed5-4198-8c93-52310dcf979c.

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

## Acknowledgements
The authors thank Dr. Natacha Aguilar de Soto for the ascidian embryos. The authors thank the UK Engineering and Physical Sciences Research Council (EPSRC) (Grant numbers EP/P030017/1, EP/M000869/1 and EP/R004854/1) for funding. Work in the Somorjai lab is supported by the Wellcome Trust ISSF, the RS Macdonald Trust Charitable and EU Horizon 2020 INFRADEV "CORBEL" (Grant number 654248).

## Author contributions
Z.Y., K.D. and G.C.S. conceived and initialized the experiments, Z.Y. constructed the optical setup and acoustic transducers. Z.Y., J.N. and P.W. carried out the experiments. Y.Q. and S.C. provided support on transducer manufacturing techniques. Y.Q. performed the finite-element analysis and acoustic characterization of the acoustic setup. K.L.H.C., I.M.L.S. and D.A.L. suggested the biological experiments and provided the samples. P.W. suggested and performed the optical flow analysis. Z.Y. and K.D. wrote the paper with contributions from all authors. K.D. supervised the study.

## Additional information

**Competing interests:** Z.Y. and K.D. are inventors on a patent related to this work filed by the University of St Andrews (UK Patent Application No. 1617089.6). All other authors declare no competing interests.

**Journal peer review Information:** *Nature Communications* thanks the anonymous reviewers for their contribution to the peer review of this work. Peer reviewer reports are available.

