## [Peer Review File · Nature Communications]

Reviewers' Comments:

Reviewer #1:

Remarks to the Author:

This manuscript from Yang and colleagues describe the use of acoustic confinement and light sheet microscopy to perform high-quality imaging of embryos in vivo. While both acoustic sample confinement and light sheet microscopy have been used before to image zebrafish embryos, this is the first that these two techniques are combined to perform live imaging. Light sheet microscopy is becoming a widespread technique to image the zebrafish heart, but so far the embryos needed to be immobilized in a low-melting agarose gel before imaging. By using acoustic confinement, the authors can maintain embryos in their natural media, which is particularly convenient for applying drugs that will diffuse quickly to the animal. The authors provide an example of the applications of this technology by measuring the effects of verapamil treatments in zebrafish cardiac function.

I have read with much interest this manuscript and found this work to be very exciting. As requested by the editor, I will comment exclusively in those areas related to zebrafish cardiac imaging and will leave the specific comments about microscopy/acoustic trapping to other reviewers whose expertise would be more appropriate.

General comments

I believe that the paper is compelling and well executed and that readers from different fields will find this manuscript of significant interest. However, there are a couple of issues that the authors should address before the paper is suitable for publication:

- 1) The authors have incorrectly identified the atrium and the ventricle of the zebrafish heart. In Figure 3 and onwards, the atrium is the structure located in the left of the image, and the ventricle is on the right. In these images, the head of the animals would be towards the bottom of the picture. The authors could confirm this if they have any transmitted light imaging where they can appreciate the blood flow. I believe this is an anatomical mistake from the authors and they will need to correct their conclusions/ text/graphs accordingly. What they measure as ventricle is the atrium and vice versa.
- 2) I feel that the conclusions of the paper would be stronger if the authors could validate their system to detect also the effect of drugs that increase the heart rate (i.e., norepinephrine or caffeine).
- 3) There are a few instances where the authors should include some references to justify their claims. For example, they mention that using agarose to image micro-organisms or marine embryos is "unacceptable," but they do not show the difference between acoustic confinement and agarose confinement, neither reference any published work that describes the morphological alterations produced by agarose.

Reviewer #2:

Remarks to the Author:

The manuscript describes an interesting method for measuring biomechanical properties of small biological samples with spatial resolution approaching the cellular level, by combining acoustic-trapping-based sample positioning with light sheet microscopy. The method is exemplified by visualizing the muscle contractions and heartbeat frequencies of zebrafish larvae exposed to different concentrations of the drug Verapamil. I believe the paper meets the requirements for publication in Nat. Comm., given the below suggestions are considered and the manuscript is revised accordingly.

General:

The paper content is focused on a real-time drug-based assay with a zebrafish model. In the abstract, the stated goal of the paper is to study cardiovascular diseases. This is not reflected in

the title of the paper. I recommend to add "... for drug-based assay studies" or something similar to the paper title.

The acoustic resonator geometry is not accurately described. The authors should describe the choice of resonator (axial) length vs. diameter and radius of curvature of the transducers, as well as the frequency, and how these parameters are affecting the performance of the trap and the force gradient near the trapping spot. They also need to compare the numerical modelling with the experimental characterization shown in Suppl. Notes 7-9.

Details:

- page 2, section "Acoustic trapping": The authors compare forces from acoustic traps with forces from optical traps. However, the compared objects trapped with each method have probably different sizes (since I assume you can't trap a 0.5-mm object in an optical trap). The authors should therefore also compare the energy densities of the fields responsible for generating the forces, since the same field amplitude results in different forces on differently sized objects.

- Viability measurements performed over 16 and 56 h of continuous acoustic trapping (page 4): At what energy density or acoustic pressure amplitude? This needs to be quantified and linked to the pressure measurements and simulations presented in the supplementary notes. The authors also need to consider other long-term and high-pressure viability studies of relevance, for example Lab Chip 10, 2727-2732 (2010) and Lab Chip 15, 3341-3349 (2015).

- The scalebars are not readable in Suppl. Figs. 8 and 10. Please also add correct labels and units.

- What is the difference between the six images in Suppl. Fig. 13? Different actuation frequencies? If so, please mark the frequency for each image.

Reviewer #3:

Remarks to the Author:

In this manuscript, the authors show the capability of an acoustic trap to hold three different embryos at three different sizes and shapes, so that optical images could be collected. They show that the mass transport of compounds in the liquid medium happens more quickly than in agarose, which can be an advantage. They contend that the sizable trapping forces do not perturb the developing embryos or their physiology. Given that US is used to heat and ablate tissues, and that some labs use acoustic traps to deform cells or tissues intentionally, it seems that more than a statement is needed.

I worry that the authors are showing three systems incompletely rather than one, documenting the performance and controls needed to support their claim of non-perturbing immobilization. I would argue strongly that a complete documentation of the approach using one species is more compelling than a less complete documentation of three. In short 3 x 50% << 100%

Reviewer #1 (Remarks to the Author):

This manuscript from Yang and colleagues describe the use of acoustic confinement and light sheet microscopy to perform high-quality imaging of embryos in vivo. While both acoustic sample confinement and light sheet microscopy have been used before to image zebrafish embryos, this is the first that these two techniques are combined to perform live imaging. Light sheet microscopy is becoming a widespread technique to image the zebrafish heart, but so far the embryos needed to be immobilized in a low-melting agarose gel before imaging. By using acoustic confinement, the authors can maintain embryos in their natural media, which is particularly convenient for applying drugs that will diffuse quickly to the animal. The authors provide an example of the applications of this technology by measuring the effects of verapamil treatments in zebrafish cardiac function.

I have read with much interest this manuscript and found this work to be very exciting. As requested by the editor, I will comment exclusively in those areas related to zebrafish cardiac imaging and will leave the specific comments about microscopy/acoustic trapping to other reviewers whose expertise would be more appropriate.

General comments

I believe that the paper is compelling and well executed and that readers from different fields will find this manuscript of significant interest. However, there are a couple of issues that the authors should address before the paper is suitable for publication:

1) The authors have incorrectly identified the atrium and the ventricle of the zebrafish heart. In Figure 3 and onwards, the atrium is the structure located in the left of the image, and the ventricle is on the right. In these images, the head of the animals would be towards the bottom of the picture. The authors could confirm this if they have any transmitted light imaging where they can appreciate the blood flow. I believe this is an anatomical mistake from the authors and they will need to correct their conclusions/text/graphs accordingly. What they measure as ventricle is the atrium and vice versa.

We thank the reviewer for pointing out this important anatomical error. We have now corrected the corresponding sections accordingly. The changes are mainly located in the third paragraph of the Results Section: “Dynamic response to drug treatment”, as well as the captions of Fig.3, 4 and 5.

2) I feel that the conclusions of the paper would be stronger if the authors could validate their system to detect also the effect of drugs that increase the heart rate (i.e., norepinephrine or caffeine).

Following the Reviewer’s suggestion, we have undertaken a further investigation on the effect of norepinephrine on heart rate. To be consistent with previous experiments, we have used 2 day-post-fertilisation (dpf) zebrafish. As expected, the samples showed an increase in heart rate upon the addition of norepinephrine, and a decrease during removal. Interestingly, the response of the zebrafish to adrenergic agonists varies greatly throughout their early stages of development³⁵. We observed a mean increase in heart beat rate of 7% in 2 dpf zebrafish; however, when repeated on two 3 dpf specimens, the heart beat increase was 17%, which is consistent with previous studies³⁴. We have included the latter study in the Supplementary Note 2. Indeed, the expression of adrenergic receptor genes in 3 and 4 dpf zebrafish can be 2-fold larger than in 2 dpf zebrafish³⁵.

In the Results Section, we now include the following:

“Additionally, the experiments were repeated with 1 mM norepinephrine (NE) to illustrate increases in heart rate. Figure 5(b) shows the normalised heart rate from five 2-dpf zebrafish. Upon addition of NE, there is a gradual and consistent increase in heart rate to around 7%, and a similar drop to resting heart rate past drug wash-out. When repeated on two 3-dpf zebrafish, the heart rate increase was 17% (**Supplementary Note 2**), consistent with previous studies³⁶. Interestingly, the sensitivity of zebrafish to adrenergic agonists greatly increases in the early stages of development. For instance, the expression of adrenergic receptor genes in 3–4-dpf zebrafish can be 2-fold larger than in 2-dpf zebrafish³⁷.”

3) There are a few instances where the authors should include some references to justify their claims. For example, they mention that using agarose to image micro-organisms or marine embryos is “unacceptable,” but they do not show the difference between acoustic confinement and agarose confinement, neither reference any published work that describes the morphological alterations produced by agarose.

There are a number of situations where physical confinement with agarose and cover slips is greatly challenged. In the first instance, physical confinement may be impractical for long term studies, such as in highly motile specimens and specimens with active ciliary machinery. In these cases, the specimens must be immobilised by fixation, heavy sedation or removal of cilia, which disrupts development and precludes measurement of dynamic behaviour. Secondly, greater forces imparted by physical confinement may disrupt developmental processes. Finally, the diffusion rate of media and drugs is greatly reduced in agarose confinement. Specifically to this point, in Supplementary Note 1, we have a clear demonstration that the diffusion rate in agarose is substantially slower than in our acoustic trap, which supports our claim.

Whilst we discuss these issues in multiple sections of the manuscript, we agree that some of these claims were not well substantiated. On the suggestion of the Reviewer, we relax the claims and now provide supporting references (Treuren, W. *et al.* bioRxiv, 370478 (2018); Mitchell, A. *et al.* J. Appl. Microbiol., **83**, 76 (1997); Turner, D. *et al.* Development, **144**, 3894 (2017).) to the above points in the introduction section.

Reviewer #2 (Remarks to the Author):

The manuscript describes an interesting method for measuring biomechanical properties of small biological samples with spatial resolution approaching the cellular level, by combining acoustic-trapping-based sample positioning with light sheet microscopy. The method is exemplified by visualizing the muscle contractions and heartbeat frequencies of zebrafish larvae exposed to different concentrations of the drug Verapamil. I believe the paper meets the requirements for publication in Nat. Comm., given the below suggestions are considered and the manuscript is revised accordingly.

General:

The paper content is focused on a real-time drug-based assay with a zebrafish model. In the abstract, the stated goal of the paper is to study cardiovascular diseases. This is not reflected in the title of the paper. I recommend to add “. . . for drug-based assay studies” or something similar to the paper title.

A main aspect of our demonstration of acoustic trapping with LSFM is, indeed, cardiovascular function in zebrafish; however, the aim of such experiments was a proof-of-novelty validation that acoustic trapping in LSFM may extend the capacity over physical trapping methods. Towards this, we have additionally trapped motile specimens, and embryos with ciliary function. Further, the mechanisms of verapamil and norepinephrine (included by suggestion of Reviewer 1) are well known, thus, we do not claim to offer new insight into pharmacology beyond the demonstration of our technique. Conversely, we believe to have demonstrated that a range of specimens can be trapped and imaged using our method, which is likely to be useful across a broad range of applications in biomedicine. As such, we would prefer to keep the broader perspective of the original title.

We recognise that this was not adequately described by the abstract. Therefore, we have modified the abstract to more accurately reflect our aims. The abstract now reads:

“Contactless sample confinement would enable a whole host of new studies in developmental biology and neuroscience, in particular, when combined with long-term, wide-field optical imaging. To achieve this goal, we demonstrate a contactless acoustic gradient force trap, created via ultrasonic transducers, for sample confinement in light sheet microscopy. Our approach allows the integration of real-time environmentally controlled experiments with wide-field low photo-toxic imaging, which we demonstrate on a variety of marine animal embryos and larvae. To illustrate the key advantages of our approach, we provide quantitative data for the dynamic response of the heartbeat of zebrafish larvae to verapamil and norepinephrine, which are known to affect cardiovascular function. Optical flow analysis allows us to explore the cardiac cycle of

the zebrafish and determine the changes in contractile volume within the heart. Overcoming the restrictions of sample immobilisation and mounting can open up a broad range of studies, with real-time drug-based assays and biomechanical analyses.”

The acoustic resonator geometry is not accurately described. The authors should describe the choice of resonator (axial) length vs. diameter and radius of curvature of the transducers, as well as the frequency, and how these parameters are affecting the performance of the trap and the force gradient near the trapping spot. They also need to compare the numerical modelling with the experimental characterization shown in Suppl. Notes 7-9.

We have now included a broader description of the acoustic geometry and function to our previous description in the Online Methods “Acoustic setup”. It now reads:

“... The outer diameter of the bowl-shaped active element was 20 mm with a focusing radius of 16 mm. This resulted in an f -number ($f\#$) of 0.8, providing a good compromise between the transducer physical size and the ultrasound focal region. The 16-mm focusing radius led to a 32 mm axial length for the confocal system, allowing space for dipping in the imaging objectives. The thickness of the active element was chosen to give a resonant frequency of ~ 1.5 MHz, resulting a 1 mm wavelength (λ) of ultrasound in water and a 1.12 mm beam diameter, similarly to the size of trapping targets. The use of small $f\#$ and high frequency also ensured a tight focus and high acoustic pressure at the focal point, leading to large pressure gradient forces near the trapping spot in the confocal system. ...”

In addition, we expand on the comparison of pressure given in Suppl. Page 14:

“... the maximum acoustic pressure with 1V input in the confocal system is measured about 0.11 MPa at 1.495 MHz. Hence, the maximum acoustic pressure about 0.82 MPa was used for the cell experiments at 7.5 V input. Despite this pressure was sufficient to trap all samples presented in the work, it is much smaller than the simulation results, i.e. 1.37 MPa per 1V input at 1.519 MHz and 0.427 MPa per 1V at 1.468 MHz. The difference is mainly attributed to the uncertainty in the resonator fabrication, including the small variations of two ultrasound transducers and imperfect alignment during confocal system construction, and the electrical impedance mismatch between the confocal system (13Ω at 1.495 MHz) and the driving electronics (50Ω).”

Details:

- page 2, section “Acoustic trapping”: The authors compare forces from acoustic traps with forces from optical traps. However, the compared objects trapped with each method have probably different sizes (since I assume you can’t trap a 0.5-mm object in an optical trap). The authors should therefore also compare the energy densities of the fields responsible for generating the forces, since the same field amplitude results in different forces on differently sized objects.

It is difficult to quantify the optical forces required to trap such large objects as it is generally considered impractical, if not impossible, to do so. For a single-beam 3D trap (optical tweezers) a “rule-of-thumb” for the force that can be generated is 1 pN for every 10 mW laser power applied (Neuman, *Rev. Sci. Instrum.* **75**(9), 2787-2809 (2004), ref 14)). Scaling this up to the 5 μ N required to trap the 500 μ m diameter glass sphere used as an example in the “Acoustic trapping” Results section of the manuscript results in a laser power of about 50 kW in a tightly focused (1 μ m diameter) beam. If instead we look at counter-propagating optical traps which are more favourable for trapping of larger objects, forces of 135 pN can be achieved with 500 mW laser power (Thalhammer, *J. Opt.* **13**(4), 044024 (2011)). Again, for the example given in the manuscript, this would require 1.85 kW laser power in a ~ 1 mm diameter beam to achieve the required forces. This results in the highly impractical optical intensity at the trap location of approximately 600 MW/cm². Additionally, Thalhammer *et al* 2011 reported that the use of 500 mW laser power resulted in a local temperature increase of 10 degrees which already precludes a large range of biological experiments. In this study, The calculated acoustic energy density is 75.48 J/m³, based on maximum pressure 0.822 MPa at 7.5 V input.

We have integrated the above discussion into Supplementary Note 8 on Page 17.

- Viability measurements performed over 16 and 56 h of continuous acoustic trapping (page 4): At what energy density or acoustic pressure amplitude? This needs to be quantified and linked to the pressure measurements and simulations presented in the supplementary notes. The authors also need to consider other long-term and high-pressure viability studies of relevance, for example Lab Chip 10, 2727-2732 (2010) and Lab Chip 15, 3341-3349 (2015).

For the continuous viability study, the maximum input 7.5V was used, corresponding to acoustic pressure amplitude of 0.822 MPa and peak acoustic intensity of 22.58 W/cm². This is in line with the literature that the reviewer has referred to. We now include these studies into the discussion in the second paragraph of discussion section. Now it reads:

The intensity of ultrasound required for trapping has been proven through a number of studies to cause little harm to biological samples. Some of those tests were performed on organisms over relatively short exposure times^{17–20} whilst others were performed on mammalian cells with MPa ultrasound pressure over longer periods of time up to days^{38–40}. Here, we have performed a long-term viability study with zebrafish larvae, as a step towards improving our understanding of the effects of long-term acoustic trapping upon a range of marine organisms. The maximum input voltage of 7.50 V was used, corresponding to acoustic pressure amplitude of 0.82 MPa and peak acoustic intensity of 22.58 W cm⁻². This is in line with other long-term cell viability studies in the literature^{39,40}, and the intensity is much less than that used in the focused ultrasound therapy⁴¹.

- The scalebars are not readable in Suppl. Figs. 8 and 10. Please also add correct labels and units.

The figures have been updated accordingly.

- What is the difference between the six images in Suppl. Fig. 13? Different actuation frequencies? If so, please mark the frequency for each image.

The six images do have varying frequencies. The frequencies were selected at random to demonstrate the capability of the Schlieren setup to capture different acoustic interference patterns and, therefore, the capacity to use such imaging to determine the precise frequency at which trapping can be achieved. The trapping frequency is sensitive to the manufactured specifications of each device, thus, each frequency is only representative. We have updated the caption of the figure (now Suppl. Fig. 16) to note that the frequencies are randomly selected.

Reviewer #3 (Remarks to the Author):

In this manuscript, the authors show the capability of an acoustic trap to hold three different embryos at three different sizes and shapes, so that optical images could be collected. They show that the mass transport of compounds in the liquid medium happens more quickly than in agarose, which can be an advantage. They contend that the sizable trapping forces do not perturb the developing embryos or their physiology. Given that US is used to heat and ablate tissues, and that some labs use acoustic traps to deform cells or tissues intentionally, it seems that more than a statement is needed.

In heating and ablating tissues with focused ultrasound, the peak intensities at the focal point generally would exceed 1000 W/cm² [G. Ter Haar, *Ultrasound Obstet Gynecol*, **32**(5), 2008.], whilst in our confocal system, the peak intensity at the focal zone was 22.58 W/cm² at 7.5 V input, which is in line with other long-term cell viability studies in literature. We now have updated our second graph in the Discussion section which now reads:

“The intensity of ultrasound required for trapping has been proven through a number of studies to cause little harm to biological samples. Some of those tests were performed on organisms over relatively short exposure times^{17–20} whilst others were performed on mammalian cells with MPa ultrasound pressure over longer periods of time up to days^{38–40}. Here, we have performed a long-term viability study with zebrafish larvae, as a step towards improving our understanding

of the effects of long-term acoustic trapping upon a range of marine organisms. The maximum input voltage of 7.50 V was used, corresponding to acoustic pressure amplitude of 0.82 MPa and peak acoustic intensity of 22.58 W cm^{-2} . This is in line with other long-term cell viability studies in literature^{39,40}, and the intensity is much less than that used in the focused ultrasound therapy⁴¹. Our findings are that, even for prolonged acoustic exposures, no adverse effects have been observed.”

Furthermore, our confocal system formed a quasi-standing wave field, where the targets were acoustically moved away from energy density maxima and trapped at the energy density minima. In trapping and deforming single cells performed by other groups (for instance: J.Y. Hwang et al, *Sci. Rep.* **6**, 27238, (2016).), the cells were trapped and deformed at the acoustic beam focus, i.e., energy density maxima. We have now made this point clear in “Acoustic setup” in Methods:

“... The confocal system formed a quasi-standing wave field, where the dense objects were acoustically moved away from energy density maxima (pressure anti-node) and trapped at the energy density minima (pressure node). ...”

I worry that the authors are showing three systems incompletely rather than one, documenting the performance and controls needed to support their claim of non-perturbing immobilization. I would argue strongly that a complete documentation of the approach using one species is more compelling than a less complete documentation of three. In short $3 \times 50\% \ll 100\%$.

In this manuscript, we present novel confinement in a light-sheet geometry as the primary goal. As such, we have sought to demonstrate the broader applicability of acoustic trapping in a number of samples that pose a challenge to traditional physical confinement methods. We believe that the demonstration of trapping of a variety of specimens is performed in good detail and broadens the interest for our proposed technique. We detail the applicability of this method on the drug response in zebrafish, in particular, which we explore to a greater depth than other samples. In response to Reviewer 1, we have now included a further study on the response to norepinephrine, which alternatively increases the heart rate. Whilst it would indeed be desirable to provide a ‘complete documentation of one species’, we believe that such work would warrant a separate consideration and publication, following on from this present paper whose focus is more on the technological aspect.

Advanced geometries for light sheet with representative biological studies was performed in previous publications in *Nature Communications*, such as:

- Medeiros, G. de *et al.* Confocal multiview light-sheet microscopy. *Nature Communications* **6**, 8881 (2015).
- Jahr, W. *et al.* Hyperspectral light sheet microscopy. *Nature Communications* **6**, 7990 (2015).
- Gustavsson *et al.* 3D single-molecule super-resolution microscopy with a tilted light sheet. *Nature Communications* **9**, 123 (2018)

Thus we see our work in this manner and indeed will follow up with detailed studies on a specific species in future.

Reviewers' Comments:

Reviewer #1:

Remarks to the Author:

The authors have addressed all my points and comments. I'm therefore very satisfied with the revision and recommend with enthusiasm the publication of this interesting work.

Reviewer #2:

Remarks to the Author:

I have carefully read the author response to my review report (reviewer 2), and the corresponding parts of the revised manuscript. I am very please to find that the authors have carefully considered all my suggestions and managed to provide in my view a much more interesting, accurate and high-quality paper in this revised version. I find the additions very interesting and thoughtful. I would like to recommend acceptance of the revised version of the manuscript.

Reviewer #3:

None